# Stable Single-Pixel Contrastive Learning for Semantic and Geometric Tasks

**Leonid Pogorelyuk**
Rensselaer Polytechnic Institute
Troy, NY, USA
`pogorl@rpi.edu`

**Niels Bracher**
Rensselaer Polytechnic Institute
Troy, NY, USA
`brachn@rpi.edu`

**Aaron Verkleeren**
Rensselaer Polytechnic Institute
Troy, NY, USA
`verkla@rpi.edu`

**Lars Kühmichel**
Technical University Dortmund
Dortmund, Germany
`lars.kuehmichel@tu-dortmund.de`

**Stefan T. Radev**
Rensselaer Polytechnic Institute
Troy, NY, USA
`radevs@rpi.edu`

## Abstract

We pilot a family of stable contrastive losses for learning pixel-level representations that jointly capture semantic and geometric information. Our approach maps each pixel of an image to an overcomplete descriptor that is both view-invariant and semantically meaningful. It enables precise point-correspondence across images without requiring momentum-based teacher–student training. Two experiments in synthetic 2D and 3D environments demonstrate the properties of our loss and the resulting overcomplete representations.

## 1 Introduction

Many modern computer vision tasks fall into geometric or semantic domains. Geometric tasks focus on the spatial aspects of images, such as matching points between two images [1, 2], camera pose estimation [3], depth estimation [4], simultaneous localization and mapping [SLAM; 5], and 3D reconstruction [6]. When such frameworks produce pixel-level features useful for spatial purposes, these features are not expected to carry semantic information about the objects in the image. Semantic tasks, on the other hand, focus on the contextual aspects such as object detection and classification [7, 8], semantic and instance segmentation [9, 10], question answering [11], or pose estimation [12]. However, when these frameworks produce features, they are at best at patch-level, not pixel-level precision. Hence, they are often too imprecise for the aforementioned geometric tasks.

In the following, we develop a novel family of geometric losses that encourage a self-supervised learner to encode semantic and view-invariant information with pixel-level precision. Encoding features with $D > 3$ dimensions per pixel can be regarded as learning overcomplete representations [13]. Such representations exhibit favorable properties across different domains and tasks. For instance, they have been applied to unpaired shape transformations in high-dimensional point clouds [14], deep subspace clustering [15], and denoising of high-dimensional images [16]. Building on this idea, our proposed loss family induces networks that can transform images into sets of overcomplete

39th Conference on Neural Information Processing Systems (NeurIPS 2025) Workshop: UniReps: Unifying Representations in Neural Models.

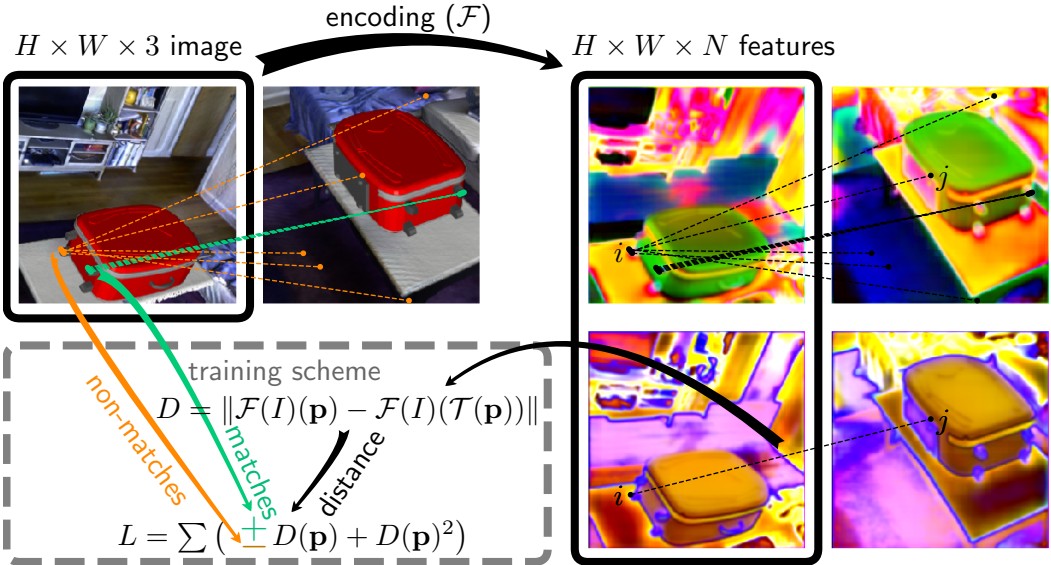

Figure 1: An experiment of matching pixels between different views (top left) of synthetic scenes with objects from ShapeNetCore [17] placed in rooms from SceneNN [18] (from **Experiment 2**). Our loss (with the $\infty$ norm) encourages non-matching features to differ by at least one channel, producing a network that encodes separate objects (instance segmentation; top right channels) and also identifies edges and interiors (bottom right), without explicitly being trained to do so.

descriptors containing both semantic and geometric information. Such descriptors can be made invariant to view changes and occlusion 3D. In sum, our contributions are:

- A new loss family for stable contrastive learning of pixel-level representations without momentum-based update schemes such as teacher-student.
- A demonstration that the loss encourages learning of semantic representations at the pixel level that are precise enough to match features across images.
- An extension of pixel-level contrastive learning to 3D environments.

## 2   Method

**Training setup.**   During training, we generate two views of the same image. Each view has pixel coordinates $\mathbf{p} \in \mathbb{N}^2$, where $\mathbf{I}(\mathbf{p}) \in \mathbb{R}^3$ is the RGB value at $\mathbf{p}$. A feature extractor $\mathcal{F}$ maps each image $\mathbf{I} \in \mathbb{R}^{H \times W \times 3}$ to an overcomplete feature map in $\mathbb{R}^{H \times W \times D}$ with $D > 3$. We use a U-Net–style encoder–decoder topology [19] with residual blocks [20] and $D$ output channels (see **Appendix A** for details); self-attention is disabled in all reported runs [21], as it did not improve performance. The transformation set $\mathbb{T}$ has geometric and photometric components. In 2D, the geometric part is a perspective transform with random rotation, scaling, and skewing; in 3D, it amounts to random camera repositioning and rotation. Photometric changes include hue/saturation jitter and, optionally, more complex noise such as motion blur [22]. Since the transformation is known during training, we have a mapping $\mathcal{T} : \mathbb{N}^2 \to \mathbb{N}^2$ that sends pixel $\mathbf{p}$ in $\mathcal{F}(\mathbf{I}_1)$ to pixel $\mathcal{T}(\mathbf{p})$ in $\mathcal{F}(\mathbf{I}_2)$.

**Within-image contrastive loss.**   Our loss consists of a *within-image* contrastive and a *between-image* contrastive component. The within-image component decomposes into a *positive* and a *negative* term. The positive term computes the $p$-norm between the feature map of one view $\mathcal{F}(\mathbf{I}_1)$ at coordinates $\mathbf{p}$ and the feature map $\mathcal{F}(\mathbf{I}_2)$ of the other view at the transformed coordinates $\mathcal{T}(\mathbf{p})$ (random coordinates $\mathcal{R}(\mathbf{p})$ are used instead for the negative term):

$$D_{\text{pos}}(\mathbf{p}) := \|\mathcal{F}(\mathbf{I}_1)(\mathbf{p}) - \mathcal{F}(\mathbf{I}_2)(\mathcal{T}(\mathbf{p}))\|_p \; ; \quad D_{\text{neg}}(\mathbf{p}) := \|\mathcal{F}(\mathbf{I}_1)(\mathbf{p}) - \mathcal{F}(\mathbf{I}_2)(\mathcal{R}(\mathbf{p}))\|_p . \quad (1)$$

We combine the two terms and aggregate them across the set of all pixels $S = P \cup N$:

$$\mathcal{L}_{\text{within}} := \frac{1}{|P|} \sum_{\mathbf{p} \in P} \left[ D_{\text{pos}}(\mathbf{p}) + D_{\text{pos}}(\mathbf{p})^2 \right] + \frac{1}{|N|} \sum_{\mathbf{p} \in N} \left[ -D_{\text{neg}}(\mathbf{p}) + D_{\text{neg}}(\mathbf{p})^2 \right], \tag{2}$$

where $P$ is the subset of "positive" (i.e., transformed) pixels and $N$ is the subset of "negative" (i.e., randomly sampled) pixels. The negative sign on the right is meant to increase the difference (in feature-space) between "negative" matches, while the square term is meant to ensure that this difference doesn't grow unbounded. The fraction of pixels $f = |P| / |S|$ designated as positive is a hyperparameter that determines the balance between positive and negative examples.

**Between-image contrastive loss.** The second component compares the overcomplete features of two different, unrelated views $\mathbf{I}$ and $\mathbf{I}'$:

$$C(\mathbf{p}) := \left\| \mathcal{F}(\mathbf{I})(\mathbf{p}) - \mathcal{F}(\mathbf{I}')(\mathbf{p}) \right\|_p \tag{3}$$

The $p$-norms are then summed over the set of all pixels $S$:

$$\mathcal{L}_{\text{between}} := \frac{1}{|S|} \sum_{\mathbf{p} \in S} \left[ -C(\mathbf{p}) + C(\mathbf{p})^2 \right], \tag{4}$$

where the views can be either the original images $\mathbf{I}_1$ or their transformed versions $\mathbf{I}_2$. We use the latter, since the feature extractor sees augmented versions of the "negative examples" in each epoch.

**Total loss.** The feature extractor minimizes the following combined loss:

$$\mathcal{L}_{\text{total}} = \lambda \cdot \mathcal{L}_{\text{within}} + (1 - \lambda) \cdot \mathcal{L}_{\text{between}}, \tag{5}$$

where $\lambda$ is a tunable hyperparameter, and the loss is taken in expectation over an unlabeled data set, the set of transformations $\mathbb{T}$, and the fraction $f$ of positive pixels in the within-image component. Ideally, once trained, the feature extractor transforms an image into a set of overcomplete descriptors containing semantic and geometric information at the pixel level.

## 3  Related Work

Large transformer-based vision models, such as ViT [23] and Swin Transformer [24], produce patch-level descriptors. Supervised methods like SAM [25] generate semantic, localized features via labeled objectives, while self-supervised learning (SSL) approaches [26–28] achieve similar semantic density without labels [29, 30]. However, due to their patch-based design, such models tend to prioritize semantic representation over spatial precision, especially when trained with contrastive losses [31], leading to insufficient localization for geometric tasks like SfM. This limitation can be mitigated in a task-specific manner by adding spatially consistent objectives during training or fine-tuning [32, 33], or by adapting ViTs for sparse keypoint detection [34].

In 2D, Xie et al. [35] showed that a geometric contrastive loss as a pretext task can learn pixel-level dense descriptors useful for downstream semantic tasks, using a momentum-based update to stabilize training—a common challenge in contrastive methods. Stojanov et al. [36] extended this idea to 3D, though at a patch rather than pixel level. The Universal Correspondence Network [UCN; 37] was notable among early descriptor-based methods for encoding per-pixel keypoints with a 3D loss function, though it remained limited to purely geometric, non-semantic learning objectives. Our approach avoids momentum-based update rules [35] with a new loss and can use both 2D and 3D data to encode semantic information and view invariance at the pixel level.

## 4  Experiments

### 4.1  Experiment 1: Domain generalization and color invariance

**Motivation and setup.** We first evaluate our loss in a classic domain generalization setting using the Colored MNIST benchmark [38]. The task requires predicting if a digit is $\geq 5$, making it a semantic task designed to separate invariant features (digit identity) from spurious ones (digit color). The training set consists of multiple domains with fixed digit–color correlations, while the test set contains unseen digit–color combinations. We vary hyperparameters $\lambda$ (trade-off between within- and between-image terms) and $p \in \ell_1, \ell_2, \ell_\infty$ (feature distance norm).

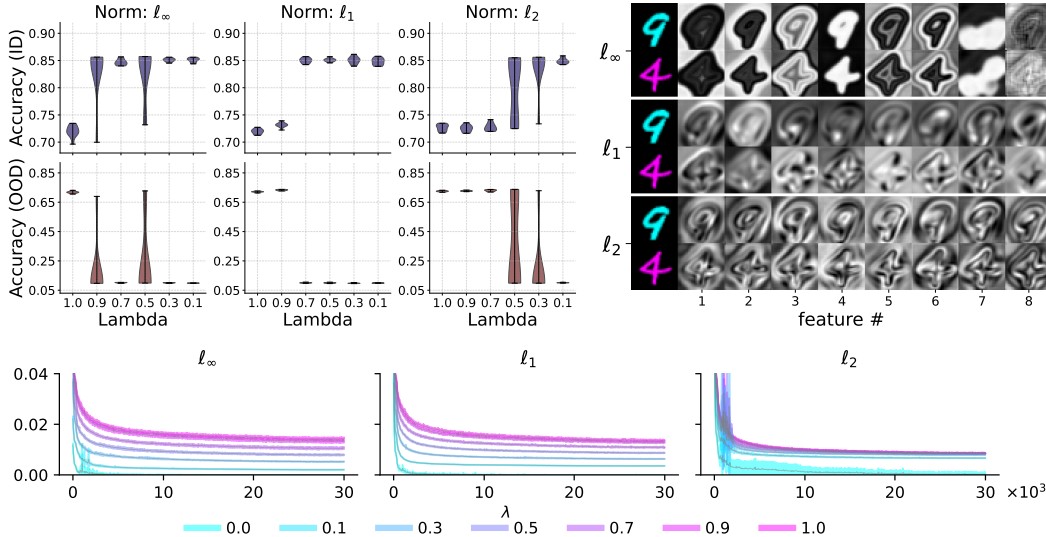

Figure 2: **Experiment 1**. *Top Left*: ID and OOD accuracy vs. $\lambda$ (relative weight) and $p$ (norm). Wide violins show seed variability: runs either encode color ($\approx$85% ID, $\approx$10% OOD) or suppress it (balanced $\approx$75%). Lower $\lambda$ (more weight on $\mathcal{L}_{\text{between}}$) increases the probability of encoding color; at $\lambda = 1$ ($\mathcal{L}_{\text{within}}$ only), features become invariant to color. *Top Right*: Feature maps from frozen backbones at $\lambda = 0.5$ for two example inputs: $\ell_\infty$ yields sharp, bit-like encodings; $\ell_1, \ell_2$ yield smoother encodings. Downstream classification on ColoredMNIST revealed color-encoding of all three backbones, which is clearly apparent for $\ell_\infty$ and $\ell_1$; however, this is not the case for $\ell_2$. *Bottom*: Shown are the loss curves for each norm averaged over the 10 seeds per $\lambda$. For $\ell_\infty$ and $\ell_1$ backbone training is stable and robust across different $\lambda$ in contrast to traditional $\ell_2$ norm. A higher weight of contrastive loss between embeddings (lower $\lambda$) results in a higher overall loss.

After training, we freeze the backbone and fit an attention-based pixel classifier averaging *pixel-level* predictions. This *permutation-invariant* classifier tests whether representations capture semantic information at the pixel level. We hypothesize that **(1)** $\lambda = 1$ (within-image loss only) will encourage color invariance, improving OOD performance; **(2)** lower $\lambda$ values will allow encoding of color, reducing OOD accuracy; and **(3)** the $\ell_\infty$ norm will produce bit-like, sharp representations, while $\ell_2$ and $\ell_1$ norms will yield smoother, averaged features.

**Results.** The results confirm that **(a)** semantic information is encoded at the pixel level; **(b)** $\lambda = 1$ produces color-invariant features while lower $\lambda$ causes the model to encode color (see Figure 2, *left*); **(c)** the $\ell_\infty$ norm yields sharp features, whereas $\ell_2$ and $\ell_1$ norms produce smoother features (see Figure 2, *right*); and **(d)** the features are localized: when they match across different images, they correspond to the same part of the object (see Figure 4). Further results are available in Appendix B.



Figure 3: Top row: source image, its transformed view (target 1), and a different image (target 2). Bottom row: Each source pixel (colored for visualization only) is mapped onto the location of the most similar (in $l_2$ distance) encoding.

## 4.2 Experiment 2: 3D invariance in synthetic scenes

**Motivation and setup.** Overcomplete representations can also be rendered invariant to viewing angle and occlusions. This requires a camera to change its position in 3D, as opposed to applying a perspective transform on the image. We illustrate this concept using synthetic scenes in which 40336 objects from ShapeNetCore [17] are placed in 76 rooms from SceneNN [18] (see Figure 1, top left).

With 3D meshes of all rooms and objects, we can randomly position and orient objects, place and rotate the camera at random, and track the visible pixels in the images. For training, we generate different views of the same room, with some objects shared across scenes and others not. About 1/8 of the pixels in one view are traced to their nearest 3D mesh intersection and matched to the corresponding pixels in the second view. Occlusions are detected by tracing back from the second view and measuring distances from the original intersections; large discrepancies indicate non-matches. The remaining non-matching pixels are chosen randomly.

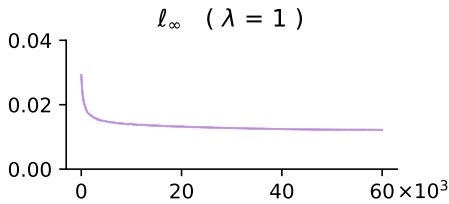

Figure 4: Validation loss curve with $\ell_\infty$ and $\lambda = 1$. The loss demonstrates stable convergence with minimal variance.

**Results.** Our loss (Eq. 5) using the $l_\infty$ norm and $\lambda = 1$ encourages the network to learn at least two modes of information in a bit-like fashion across its channels: *Semantic*—different objects/parts have encoding different by at least one channel but consistent across views, akin to instance segmentation (Figure 1, top right); *Geometric*—some of the channels flip sign depending on how far a point is from an edge, with some channels effectively detect shape edges and other detect "inner" regions in 3D (Figure 1, bottom right). Further results are available in Appendix C.

## 5    Limitations and Conclusion

We piloted a new family of losses for stable contrastive learning of pixel-level representations in 2D and 3D environments. They induce semantic information localized at the pixel level. Our approach has several limitations: overcomplete embeddings are less compute-efficient than low-dimensional ones, and artificial perspective transforms can create a simulation gap for real-world viewpoint shifts. We plan to address the latter in a series of experiments on stereo images with known point matches.

## Acknowledgments and Disclosure of Funding

This work is partially funded by the Deutsche Forschungsgemeinschaft (DFG, German Research Foundation) Project 528702768.

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

# Appendix

## A   Source code

The source code is available at `https://github.com/stefanradev93/oxels`.

## B   Details and additional results for Experiment 1

### B.1   Experimental details

Starting from MNIST ($28 \times 28$ grayscale), we binarize labels ($< 5 \rightarrow 0$, $\geq 5 \rightarrow 1$) and flip with probability 0.25. Each image's background is assigned one of two colors, sampled to correlate with the (possibly flipped) label with probability $p_{\text{train}} = 0.85$ in the training pool and $p_{\text{test}} = 0.10$ in the OOD domain. The final images are bilinearly resized to $32 \times 32$ to match the model input. Under this construction, heuristic ceilings are: shape-only $\approx 75\%$ across domains; color-only $\approx 85\%$ (ID) and $\approx 10\%$ (OOD).

### B.2   Training details and hyperparameters

All backbone and classifier models were trained on a single NVIDIA GeForce RTX 4090 with 24 GB memory using PyTorch 2.6 with CUDA 12.4. We used the AdamW optimizer [39] ($\beta_1 = 0.9$, $\beta_2 = 0.99$, weight decay $= 10^{-6}$) with a OneCycleLR schedule [40]. For the backbone trainings, the learning rate was warmed up from $4 \times 10^{-5}$ to a peak of $1 \times 10^{-3}$ over the first 5% of training steps, then cosine-annealed to $1 \times 10^{-7}$ over the remainder. The classifier trainings followed a similar regime with a peak learning rate of $2 \times 10^{-3}$.

Training proceeded sequentially: we first trained each backbone, then trained a classifier with the backbone frozen. For the $D$-sweep with $D \in \{4, 8, 16, 32\}$ at $\lambda = 1.0$, and for the additional five runs per norm at $\lambda = 0.1$ with $D = 32$, each run used 100,000 steps for the backbone training and 30,000 steps for the classifier. A 100,000-step run took $\sim$1.5 hours on the RTX 4090; scaling this to the 210-run $\lambda$-sweep would have required $\sim$315 GPU-hours for the backbones alone. To make the sweep feasible, we reduced the per-run budget of the backbones to 30,000 steps ($\sim$0.45 hours/run) and the classifiers to 12,000, accepting a modest loss-convergence degradation for broader coverage. The $\lambda$-sweep (seven values under $\ell_\infty$, $\ell_1$, and $\ell_2$ with 10 seeds each, totalling 210 runs) therefore used 30,000/12,000 steps per run (backbone/classifier).

### B.3   Backbone architectures

We use a U-Net variant following [21] (without time embeddings). An initial $3 \times 3$ convolution maps RGB to 32 channels, followed by three encoder stages with 1, 2, and 2 residual blocks at widths 32, 64, and 64. Blocks use SiLU and a LayerNorm-style normalization; residual dropout $p = 0.1$ is applied only in the last two stages. Each stage downsamples by $\times \frac{1}{2}$ via average pooling with a $1 \times 1$ convolution for channel alignment. The bottleneck is a residual block $\rightarrow$ self-attention $\rightarrow$ residual block (attention only here). The decoder mirrors the encoder, upsampling by $\times 2$ using a $1 \times 1$ convolution followed by nearest-neighbor interpolation, with one extra residual block per stage to fuse skip connections (the last block has no skip). The head is normalization $\rightarrow$ SiLU $\rightarrow$ $3 \times 3$ convolution projecting to $D$ output channels (we use $D=32$ unless varied). Convolutions are initialized with Xavier uniform and near-zero biases. Each backbone has a total of 993,664 parameters.

### B.4   Classifier architectures

We use a single-head self-attention pixel classifier (similar to the self-attention block in [21]) on the backbone features $\mathbf{F} \in \mathbb{R}^{H \times W \times D}$. Inputs are per-pixel normalized with a LayerNorm-style scheme (one-group normalization; no bias shift). Linear projections produce $Q, K, V \in \mathbb{R}^D$; $Q$ and $K$ are normalized and scaled by $1/\sqrt{D}$. Attention is computed over all pixels (no positional encodings), yielding attended pixel embeddings that a small MLP (hidden widths 150, 100, 50) maps to a *single* logit per pixel (binary task). The image-level prediction is the uniform average of pixel logits. The backbone remains frozen, and a classifier with 28,479 parameters is trained for each backbone.

## B.5 Additional results

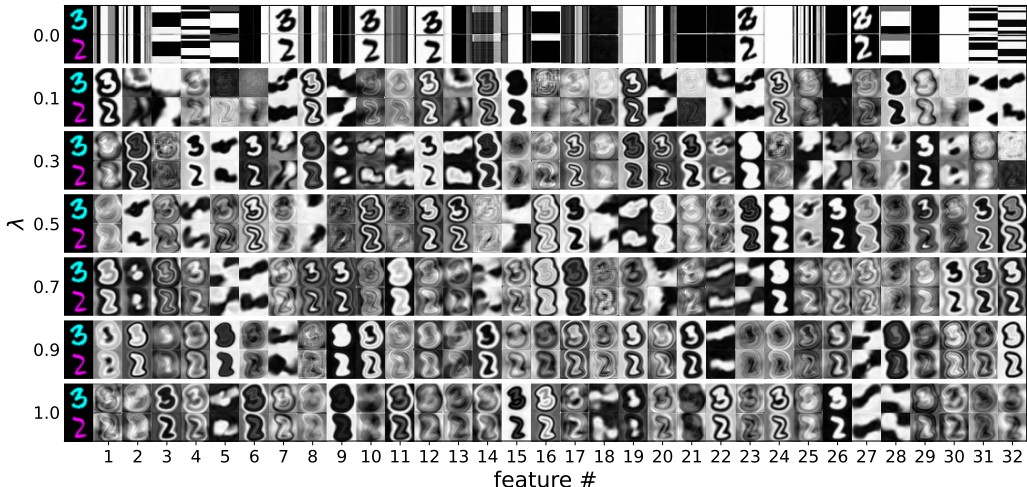

Figure 5: Example overcomplete representations obtained via an $l_\infty$-based loss from validation samples.

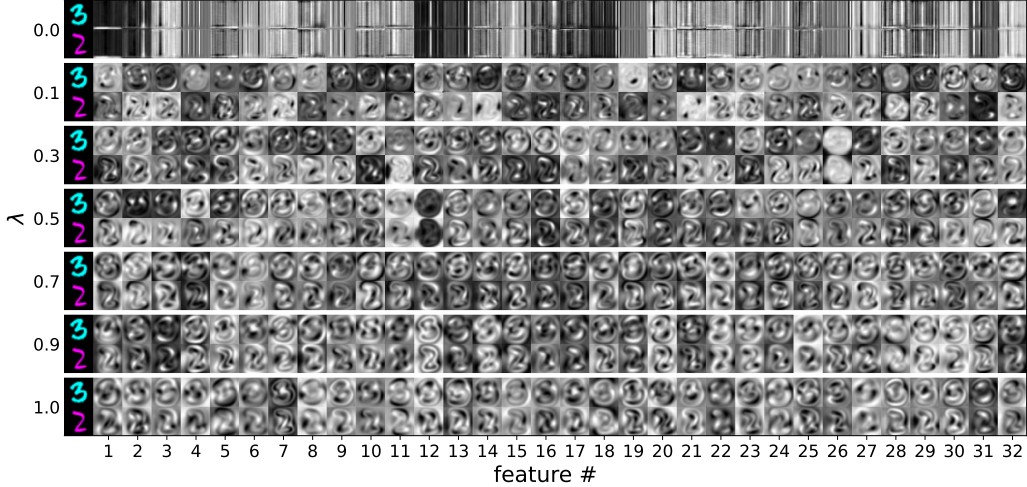

Figure 6: Example overcomplete representations obtained via an $l_1$-based loss from validation samples.

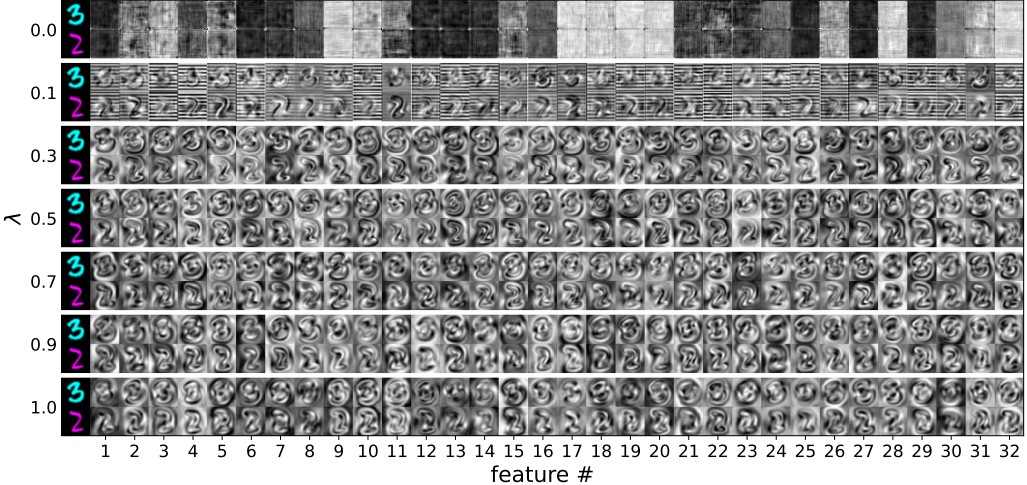

Figure 7: Example overcomplete representations obtained via an $l_2$-based loss from validation samples.

## C   Details and additional results for Experiment 2

### C.1   Training details hyperparameters

All models were trained using an NVIDIA RTX 4090 GPU with 24 GB memory on PyTorch 2.6 with CUDA 12.4. We used the AdamW optimizer with a OneCycle schedule. The initial learning rate starts at 4.0e-5, warms up to 1.0e-3 over the first 5% of steps, then anneals to 4.0e-9 by the end. Training ran for 6 epochs with a batch size of 32 for a total of 4 hours.

### C.2   Backbone architecture

The backbone was a U-Net with residual blocks, producing 32 output channels. It has five stages with channel widths [16, 16, 32, 32, 64] and residual block counts [1, 1, 1, 2, 4]. Dropout with $p$=0.1 is applied on the last two stages. The backbone had a total of 1,031,344 parameters.

### C.3   Additional results

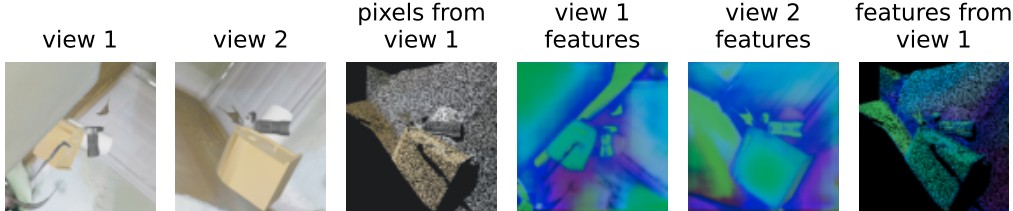

Figure 8: **Training example in 3D:** Training data pairs consisted of two views of a 3D scene, as well as a known transformation, $\mathcal{T}$, that maps pixels from view 1 to match their corresponding locations in view 2. Our loss function (Equation 2 with $\lambda = 1$ and $p = \infty$) uses $\mathcal{T}$ to compare the network-extracted features (overcomplete representations) of the two views at a fraction of the matching pixels (positive pairs) and at random pixels (negative pairs). For visualization purposes, we show just the first three channels of the features, scaled by 0.5 and shifted by 0.5 to form "false" color images.

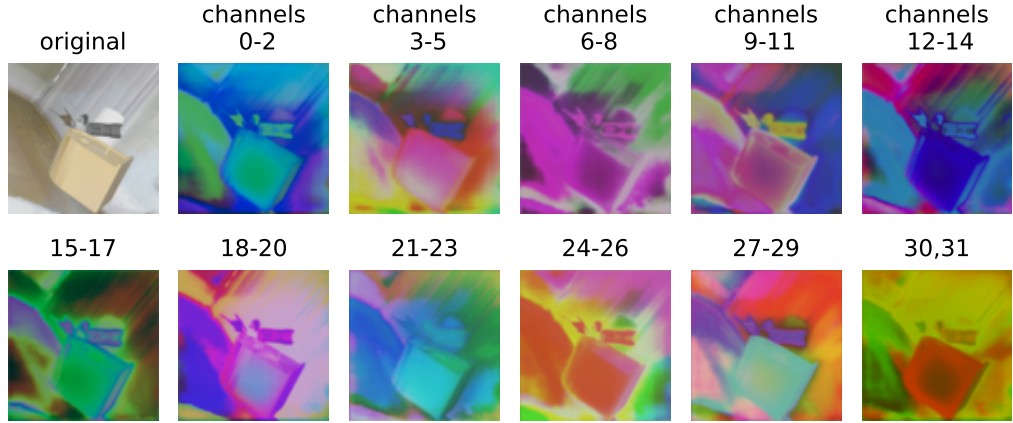

Figure 9: Example of overcomplete representations for all the channels encoded by the network, trained with $\lambda = 1$ and $p = \infty$ on our synthetic 3D data.

## D   Societal Impact

Since this paper focuses on foundational research and methodology, we expect that there is no direct path for negative societal impact. In fact, positive impacts may manifest, for instance, in improved performance or robustness of good-faith downstream applications.

Nevertheless, we cannot rule out intentional or unintentional misuse of our methods for harmful (downstream) applications.

