# OpenReview forum: "Stable Single-Pixel Contrastive Learning for Semantic and Geometric Tasks"
_NeurIPS.cc/2025/Workshop/UniReps — UniReps2025_

### Official Review · Reviewer_qajW · 2025-09-08
**Review of Stable Single-Pixel Contrastive Learning for Semantic and Geometric Tasks**

**Confidence:** 5

**Review:**

Summary
     This paper aims to learn a robust pixel level representation that is view invariant and semantically meaningful. To implement this, the loss encourages a pixel at different viewpoints/images from the same 3D point has similar embedding. Similarly, different pixels from different 3D point should have different embedding.

Strength:
Figure 1 is clear and help the user to understand the proposed method. Section2 is also simple to follow.

Weakness.
1.	Both the main paper and the appendix did not provide any qualitative comparison of the model performance. There is no baseline as well
2.	This paper and its loss is similar to paper like PointContrast[a] and should show some performance on dataset like ScanNet. Both are missing in the manuscript.
3.	It is unclear why the paper show result on colored MNIST, which is not a representative dataset for evaluate the proposed framework.
4.	Given that there is D_{pos} in Eq2, why there is an addition D_{pos}^2 (p) term in equation 2? Is there any ablation study about why the term D_{pos}^2 (p) is needed?

**Score:**

1

**Topic Fit:**

2

---

### Official Review · Reviewer_xY75 · 2025-09-13
**The goal of adding geometric information to features to enhance spatial understanding is timely and needed, the experiments are only on MNIST which is not enough, the loss function is not clear to me**

**Confidence:** 4

**Review:**

Summary:

The papers aims to find pixel-level representations that are semantic and spatially consistent. For instance, the pixels of a head of a dog should have similar features in all images, which is what current foundation models also need to have to get better spatial understanding.

Strenghts:

1 - The goal of the paper is timely, knowing that spatial understanding is a limitation in current foundation models.

Weaknesses:

1 - The proposed loss function is confusing. In Eq1&2 I don't understand why the square of each term is needed, and furthermore, why it has a positive sign for negative coordinates. Isn't it as opposed to what we want to do, which is making the representation of pixels that com from the same transformations similar while far away from random pixels?

2 - The experiment is only done on MNIST,  which makes the practicality of the loss function questionable. I think more experiments on COCO and PASCAL  should be done.

3 - The related work is missing some prominent works such as Leopart[1] and Timetuning[2], which follow a very similar intention.

[1] - Ziegler, Adrian, and Yuki M. Asano. "Self-supervised learning of object parts for semantic segmentation." Proceedings of the IEEE/CVF conference on computer vision and pattern recognition. 2022.

[2] - Salehi, Mohammadreza, et al. "Time does tell: Self-supervised time-tuning of dense image representations." Proceedings of the IEEE/CVF International Conference on Computer Vision. 2023.

**Score:**

1

**Topic Fit:**

2

---

### Official Review · Reviewer_bA1W · 2025-09-15
**A preliminary study with a novel pixel-level contrastive loss**

**Confidence:** 4

**Review:**

**Summary:**

The paper introduces a new pixel-level contrastive loss for self-supervised pre-training without using a teacher-student design or momentum updates.

**Strengths:**

1. The pixel-level loss is novel and different from previous contrastive losses.
2. The feature visualizations in Figure 1 and in the Appendix show that semantic and geometric information can be learned with that loss.
3. The paper is logically decomposed and can be understood easily. Figure 1 provides a good overview of the loss and the method.

**Major Weaknesses:**

1. The evaluation is limited to small-scale (Colored MNIST) and synthetic 3D data.
2. There are no comparisons to baselines or other loss functions.
3. Experiment 2 is evaluated purely qualitatively. Adding some quantitative evaluations on semantic and geometric downstream tasks could emphasize the results even more.

**Minor Weaknesses:**

1. The method is very similar to [1] but for images/3D data instead of videos. To the best of my knowledge, [1] was not published at a conference yet, but it would still make sense to reference the paper when introducing the loss. Also, some intuition behind the loss could be helpful.
2. The title and the abstract mentions that this loss formulation is stable. This claim could be backed by showing loss curves and comparing the loss to a less stable variant.

**Justification For Recommendation:**

The rating of accept is mainly based on the novelty of the loss function and the clarity of the paper. Even though the evaluation could be improved, the paper introduces an interesting idea and some preliminary results for it.

**References:**

[1] Pogorelyuk, Leonid, and Stefan T. Radev. "Aligning motion-blurred images using contrastive learning on overcomplete pixels." arXiv preprint arXiv:2410.07410 (2024).

**Score:**

4

**Topic Fit:**

2

---

### Official Review · Reviewer_zcHe · 2025-09-18

**Confidence:** 4

**Review:**

## Summary

This paper proposes a parametric family of pixel level contrastive losses for learning per pixel embeddings that capture both semantic and geometric information. The losses have two components: (1) a within-image term that pulls together embeddings of corresponding pixels under known transformations and pushes apart embeddings of randomly paired pixels, and (2) a between-image term that separates pixels from unrelated images. Experiments on ColoredMNIST show that the method controls whether color is encoded or suppressed, and that different $\ell_p$ norms yield smoother or more discrete embeddings. Experiments on synthetic 3D scenes demonstrate that the learned embeddings separate objects and capture edge/interior structures across views and occlusions.

## Strengths

* Clarity: The paper is clearly written and easy to follow.
* Novelty: A simple and stable pixel-level contrastive framework without teacher–student mechanisms.
* Contribution: Shows that pixel embeddings can encode both semantic and geometric cues, and that loss design choices control invariance properties. Extends pixel-level SSL to synthetic 3D scenes.
* Relevance: Fits UniReps well, as the work is concerned with semantic and geometric representation learning.

## Weaknesses

* Limited experimental scope: Results are limited to ColoredMNIST and synthetic 3D scenes. More tests on real-world datasets are required to reach a conclusion.
* Missing baselines: No baselines against existing dense/pixel-level SSL methods (e.g., DenseCL, PixPro).
* Efficiency: Overcomplete descriptors could be computationally demanding, and the trade-off is not explored.

## Suggestions (forward-looking)

* Expand experimental setup.
* Provide comparisons to other baselines.
* Run ablation tests on descriptor dimension, etc.

Lastly, a theoretical analysis of stability and norm choice would be neat.

**Score:**

4

**Topic Fit:**

3